# Peer review of "Label-Free Split Aptamer Sensor for Femtomolar Detection of Dopamine by Means of Flexible Organic Electrochemical Transistors"

_materials, 2020, doi:10.3390/ma13112577_

Round 1
Reviewer 1 Report
The authors reported novel electrochemical sensors using a splitted dopamine DNA aptamer for a sensitive and selective detection, claiming the lowest LOD of 0.5 fM.
While the strategy of splitting DNA aptamer in order to lower the background is innovative, it is not convincing if two splitted aptamer fragments form a correct quadruplex structure in the presence of its target, dopamine. Is these splitted aptamer sequences novel in this manuscript? If not, please include a corresponding reference. If it is novel, major concerns are below with some minor comments at the end.
Although author showed a data in Fig 1a that there was no signal in the absence of dopamine, they missed another very critical control experiment, which is monitoring current change with dopamine concentration variations in the absence of aptamer 2.
What is the design strategy in splitting the original ssDNA aptamer into two fragments without defecting its secondary structure which is critical for its selective dopamine binding? Does sandwiching of two aptamer fragments maintain a compatible affinity compared with the original ssDNA aptamer?
The reported Kd of original ssDNA aptamer, assuming the sequence was from reference #6, was 150 nM. What was the Kd of this sandwiched aptamer? How many dopamine molecules bind per aptamer molecule? It is unlikely that the electric current increased over 100 uM when the dopamine was subjected with aptamer 2 at constant concentration of 0.5 uM.
The reference #2 was cited to support the current drop over 117.2 uM of dopamine but I can’t see the reference is explaining dopamine polymerization at higher concentration. Dopamine dose dependency data in reference #6 showed a plateau once the assay was saturated. How does dopamine self-polymerize at higher concentration?
According to Nakatsuka, et. al., (reference #6), conformational change of ssDNA aptamer in the presence of dopamine induced moving of the substation portion of the negatively charged backbone of aptamer closer to the electrode surface, resulting charge transfer. Therefore, it should be clarified that the signal change observed in this manuscript is not due to the reorientation of immobilized aptamer 1 in the presence of dopamine, regardless of aptamer 2.
Selectivity validation: although the authors cited reference #6 to support the selectivity against structurally related catecholamines, such as norepinephrine and epinephrine, 3,4-dihydroxyphenylacetic acid (DOPAC), vanillic acid (VA), 3-methoxytyramine (3-MT), or tyramine, it should be validated in this sensor platforms because of different aptamer strategy. Sandwiching and MB labeling to a part of aptamer can affect sensitivity and/or selectivity.
Regeneration and Fig 2d: The current looks continuously increasing from the 2nd washing unlike other sensors which normally show decreasing efficiency. About 17% increase after 3 washing cycle is significant. To prove a reproducible regeneration, this experiment needs more repeats, ~10 times.
Incubation time: 10 min incubation was mention in Figure 4. Is 10 min sufficient for aptamer sandwiching? Incubation time dependency needs to be validated.
Minor corrections:
A reference for dopamine aptamer sequence should be cited.
It would be informative to readers if references stating the advantages of aptamer receptors-based biosensors such as high affinity, selectivity, low cost, and easy-fabrication are included.
Throughout the manuscript, the reference citation is not consistent. Please check the citation format.
tris-EDTA buffer: change to TE buffer
H2SO4: change to H2SO4 (numbers should be subscript)
tris: change to Tris (capital T)
Line 43, liquors: change to fluids
Line 135, targets binding events: change to target-binding events
Line 121, tris(hydroxymethyl)aminomethane buffer: change to Tris buffer
Line 182, “concentrations, Figure 2a, which may on the one hand interfered the formation of aptamer/analyte..” Figure 2a is not relevant with the description.
Lines 211-212, please cite corresponding references for clinical concentrations in each different system (adrenal chromaffin cells, human serum, and brain) instead of copying from other’s citation.
Line 238, “channel current was determined, Figure 3c [20].” Is the reference #20 relevant?
Line 301, “that the immobilization of the split aptamer2 strand together with the analyte…” Analyte and aptamer 2 were incubated not immobilized, so they can be washed out.
Reviewer 2 Report
General Comments:
In this manuscript, the authors developed an OECT based dopamine sensor, by functionalizing a gold (gate electrode) with aptamers that bind specifically to dopamine. The aptamers were additionally functionalized with a redox (methylene blue). In this way the gate electrode can be utilized as a working electrode in traditional electrochemical cells. However, when the gold electrode is used as a gate in an OECT the detection limit orders of magnitude lower reaching the highest limit of detection similar to the ones reported with EGOFETs. The authors demonstrate also good selectivity. The claims are supported by experimental data. Some minor claims need to be addressed and some minor typos should be corrected prior to publication.
Specific Comments:
iOECT is mentioned before being defined. It should be defined in line 64 and not later in experimental section.
Lines:59-61 Authors claim that previous methods include complicated processes including catalytic materials, however I believe part of their detection method includes similar complicated processes such as functionalization of methylene blue to aptamers. I believe this claim is rather misleading and undermines work from other groups and should be modified.
Minor typos with superscripts exist in the manuscript
Reviewer 3 Report
Liang et al describe in their manuscript "Label-free split aptamer sensor for femtomolar detection of dopamine by means of flexible organic electrochemical transistors" the design and characterization of aptamer based OECT sensors. The authors reach a very high resolution, which is essential for some applications of dopamine sensors. The manuscript is well written.
I suggest to take the following consideration into account before publishing the manuscript:
- At the current stage, the manuscript is rather descriptive in respect to the precise aptamer sensor mechanism. I suggest to provide a section describing the precise binding and folding mechanism of aptamer 1, aptamer 2, and dopamine. Have additional techniques been used to verify the proposed mechanism?
- Can the authors provide a control experiment for the amperometric sensors shown in Figure 1, i.e. the response of the sensors to dopamine without addition of aptamer 2?
- Can the authors comment on the response time scale of their sensors? For some applications in the neurosciences, fast sensors in the nM regime are needed. What are the time limits observed here?
- The shift in the transconductance after addition of dopamine is very small at low concentrations (i.e. only a few mV). Usually, OECTs show a hysteresis in their transfer characteristic. Can the authors provide several consecutive forward and backward scans of their transfer characteristic to be able to compare the observed voltage shift in relation to the hysteresis?
minor items:
- there are two "Figure 1"
- I suggest to only show Figure 4d, not 4c as well (seems they plot identical data)
- formatting of references is inconsistent
Reviewer 4 Report
In this manuscript, Liang and coworkers investigate on the sensing figure of merits of aptamer-functionalized gold electrodes as gating elements for organic electrochemical transistors used as amperometric sensors. The strategy has some novelty and inspires from the literature on EGOFETs by the use of a particular biological probe for functionalizing the gate, specifically applied with OECTs. The systematic approach is well made and detailed, performances are compared to the state of the art literature and fully justify publishing them as an article in MDPI’s Material journal:
Liang and coworkers have introduced their motivations and stated the literature on molecular sensing with OECTs (5 refs) before introducing their aptamer approach. The experimental section is well detailed with enough information to attempt replicating their approach. The result and discussion section is composed of a first part “PoC validation of electrochemical sensitivity of aptamer coated gold electrodes” with a sensitivity study (9 values in logC), selectivity study (5 analytes in similar conditions), reversibility study (Cycle tests); and a second part on “validating the PoC integrated on an OECT amperometric sensor” with replication of sensitivity (10 values in logC) and selectivity results (same 5 analytes as previously). The authors have tabled their results in light of the state of the art literature which justifies the choice for the title of their manuscript, before summarizing the manuscript in the conclusion.
In order to optimize the impact of their work, here below are minor suggestions that can be addressed before publishing the manuscript:
- Citing key publications on the same strategy applied for EGOFETs (not particularly OECTs) might be appropriate prior introducing OECTs and their advantages over standard EGOFETs.
- A schematic in the first figure might help to visualize the setup for integrating the aptamer functionalized wire with the OECT for the readership that are less familiar with the device aspects than the biosensing ones.
- iOECT is coined by the authors at line 76 and the new acronym explained only at line 107. Please explain the acronym ahead (I don’t recommend specifying the “interdigitated” aspect of the electrodes to create a new acronym, since it is common practice for conductimetric, amperometric and capacitive sensors).
- 107: do the authors refer by “channel area” of the whole area covered by PEDOT:PSS polymer or the area of polymer specifically between both electrodes? Since the aspects A(interelectrode)/A(total) is a parameter that impacts on the drain current modulation (Sensors 17(3) 570, 2017), it seems relevant to precise it.
- 204-205: “approximately to zero, close to the background” lacks of accuracy and avoids appreciating S/N.
- 230-231: Precise that the current is Ohmic, only until it starts to reach the saturation that clearly appears in Fig3c.
- 231-232: Please comment also on the fact that the OECT works both on accumulation and depletion mode, which is an interesting feature.
- l249: “error bar” should be replaced by “scale bar”.
Round 2
Reviewer 1 Report
The authors have tried hard to address reviewers’ comments and supplied additional information. I recommend publishing this manuscript after a few minor corrections.
There are two (e)s in Figure 5 legend. The last one should be (f).
Please refer supplemental information within the text accordingly. For example, there are no descriptions about Figures S4/S6 and Kd.
Please number supplemental figures in order. For example, Figure S5 (Line 263) was appeared before Figure S3 (Line 266).
